environmental chemistry/nanotechnology

mesoporous ceramic functional nanomaterials, cadmium, adsorption mechanism

**Authors for correspondence:**
Hongxiang Hu
e-mail: hongxianghu@163.com
Jieying Huang
e-mail: hjy@ahau.edu.cn

This article has been edited by the Royal Society of Chemistry, including the commissioning, peer review process and editorial aspects up to the point of acceptance.

# Adsorption of Cd(II) in water by mesoporous ceramic functional nanomaterials

Zhongjun Xue[1,2], Na Liu[1,2], Hongxiang Hu[1,2], Jieying Huang[1,2], Yusef Kianpoor Kalkhajeh[1,2], Xiuyuan Wu[1,2], Nian Xu[1,2], Xiaofei Fu[1,2] and Linchuan Zhan[1,2]

[1]School of Resources and Environment, Anhui Agricultural University, 230036 Hefei, People's Republic of China
[2]Anhui Province Key Laboratory of Farmland Ecological Conservation and Pollution Prevention, 230036 Hefei, People's Republic of China

ZX, 0000-0002-2571-6473; HH, 0000-0003-4987-7105; JH, 0000-0003-0694-9498

Mesoporous ceramic functional nanomaterials (MCFN) is a self-assembled environmental adsorbent with a monolayer molecular which is widely used in the treatment of industrial wastewater and contaminated soil. This work aimed to study the relationship between the adsorption behaviour of Cd(II) by MCFN and contact time, initial concentration, MCFN dosage, pH, oscillation rate and temperature through a batch adsorption method. The adsorption kinetic and isotherm behaviours were well described by the pseudo-second-order and Langmuir models. The batch characterization technique revealed that MCFN had several oxygen-containing functional groups. Using Langmuir model, the maximum adsorption capacity of MCFN for Cd(II) was 97.09 mg g$^{-1}$ at pH 6, 25°C, dosage of 0.2 g and contact time of 180 min. Thermodynamic study indicated that the present adsorption process was feasible, spontaneous and exothermic at the temperature range of 25–55°C. The results of this study provide an important enlightenment for Cd removal or preconcentration of porous ceramic nanomaterial adsorbents for environmental applications.

## 1. Introduction

In recent years, intensive application of agro-chemicals in agriculture and industry has caused discharges of heavy metals to the environment in large quantities which, in turn, can lead to harmful effects to the human health [1,2]. Some of these

heavy metals such as cadmium are potentially toxic even at very low concentrations [3]. Several conventional methods have been widely used to remove heavy metals including precipitation, ion exchange, filtration and solvent extraction [4]. However, these methods require high energy consumption of reagents, have low selectivity and high operation cost, and generate the secondary pollutants [5].

Adsorption method has been traditionally applied to remove the heavy metals [6]. However, recent innovations have developed the varieties of biological and industrial adsorbents. At present, biological adsorption is one of the most recommended methods to remediate the heavy metals' contaminated wastewater and soil [7]. Despite 'biological accumulation', biological adsorption is a mechanism of adsorption or complexation of dissolved metals based on microbial and biomass chemical activity. Compared with traditional methods, it has lower cost, higher speed and efficiency, consumes less energy. Furthermore, it involves simple operation and ecology [6]. Biological accumulation removes metals by the metabolic activity of living organisms of 'super-enriched plants'. Nevertheless, there are still shortcomings in regard to application of biological accumulation including slow speed, long cycle, low efficiency and complex selection of varieties [8].

The current studies have introduced several low-cost and abundant bioabsorbents [6,9,10] to remove cadmium from wastewater such as the adsorption of marine algae materials [6−8,11−13]. Studies have shown that alginate and fucoidan, such as brown algae, cyanobacteria and spirulina, have heavy metal chelation. Different algae have different affinity and adsorption effect on various heavy metals. Therefore, it has been proved to be effective and reliable in removing heavy metals from aqueous solutions. In addition, the crop straws (e.g. rape straw, rice straw, corn straw and wheat straw) have been successfully applied to adsorb the cadmium from wastewater due to their lignin, hemicellulose, lipid, protein, simple sugars, hydrocarbons, starch and ash components [9,10,14−25]. Moreover, crop residues have a wide variety of functional groups which react with metal cations in wastewater. However, they do not have large sorption capacity and high efficiency [16].

Industrial adsorbing materials are also among the popular heavy metal absorbents. So far, graphene, sepiolite, zeolite and clay minerals have been examined extensively [17−21]. Carbon is one of the most active and widely used adsorbents to remediate wastewater and soil due to its high specific surface area and mesoporous volume structure which can be found in mesoporous activated carbon, bamboo charcoal and biochar [22−27]. With the progress of nanometre science and engineering technology, nanomaterial adsorbents are gradually getting used to remove heavy metal removals [28].

The specific surface area of mesoporous ceramic functional nanomaterials (MCFN) can be as high as $180 \, \text{m}^2 \, \text{g}^{-1}$, providing a high adsorption rate which is necessary to adsorb and bond heavy metals. The nano-porous ceramic with a median pore of 28 nm diameter has the advantage to effectively adsorb heavy metals [29].

The objectives of this research were (i) to investigate the effect of water chemistry (e.g. contact time, pH, temperature oscillation rate and initial Cd(II) concentration) on Cd(II) adsorption onto MCFN by batch techniques; (ii) to determine the adsorption mechanism of Cd(II) on MCFN by surface complexation modelling; and (iii) to characterize the microscopic properties of MCFN by scanning electron microscopy (SEM) and Fourier transform infrared spectroscopy (FTIR). The highlight of this paper is to evaluate the potential engineering application of an efficient and inexpensive adsorbent for heavy metal immobilization in environmental purification.

# 2. Material and methods

## 2.1. Materials and instrument

MCFN was purchased from Wuhu Gefeng Co., Ltd, composed of clay minerals, kaolin, montmorillonite and phosphorus compounds. The stock Cd(II) solution ($200 \, \text{mg} \, \text{l}^{-1}$) was prepared by dissolving the anhydrous $CdCl_2$ (guaranteed reagent grade 99.99%, Aladdin Chemical, Shanghai, China). All the other reagents ($HCl$, $HNO_3$, $NaOH$, etc.) were of guaranteed grade, and used without further purification.

*Instrument*. German Analytik Jena, MPE60 flame-graphite furnace atomic absorption spectrometer, microcomputer-controlled eight-lamp holder, three-magnetic field Zeeman and deuterium hollow cathode lamp double-button background, for the determination of Cd content. American IS-RDV3 coolable constant temperature shaker, to regulate and control temperature. Type PHS-3C exact pH-meter, Japan Hitachi S-4800 scanning electron microscope (SEM), system accessories for Hitachi E-1010 and Emitech K850. USA Thermo Scientific Nicolette 50 Fourier transform infrared spectrometer

(FTIR), wavelength range 7800–350 cm$^{-1}$; the spectral resolution is better than 0.09 cm$^{-1}$ and the wavenumber accuracy is better than 0.01 cm$^{-1}$.

## 2.2. Experimental method

*Adsorption kinetics test*. CdCl$_2$ solution (50 ml), with Cd(II) that has 200 mg l$^{-1}$ concentration and pH 6, was taken. With 0.2 g MCFN added, and oscillated with the rotation speed at 180 r.p.m. under 25°C. The time gradient was set to 30, 60, 90, 120, 150, 180, 240 and 300 min. The solution was quickly taken out for filtration at a predetermined time point to determine the mass concentration of Cd(II) in the solution.

*Isothermal adsorption test*. MCFN (0.2 g), 50 ml Cd(II) solution with Cd(II) initial concentration of 50.0–500.0 mg l$^{-1}$ were, respectively, placed in a conical flask and oscillated at the rate of 180 r.p.m. at 25°C for 180 min (preliminary kinetic experiments showed that the adsorption equilibrium was achieved after 180 min). After that, the mass concentration was measured by filtration.

*Effect of adsorbent dosage on Cd(II) adsorption*. Solution (50 ml) with Cd(II) concentration of 200 mg l$^{-1}$ was taken. Portions of 0.05, 0.1, 0.15, 0.2, 0.25, 0.3, 0.4, 0.5 and 1.0 g MCFN were added to the solution and oscillated at 180 r.p.m. under 25°C, then Whatman's medium-speed 202 quantitative filter paper with a pore diameter of 20 μm was used for filtration and detection.

*Effect of oscillation rate on Cd(II) adsorption*. In 50 ml of 200 mg l$^{-1}$ Cd(II) solution, 0.2 g of MCFN was added, then oscillated for 180 min at 100, 150, 180 and 200 r.p.m. within a constant temperature (25°C), then centrifuged and Whatman's medium-speed 202 quantitative filter paper with a pore diameter of 20 μm was used for filtration and detection.

*Effect of the initial pH value of solution on Cd(II) adsorption*. MCFN (4 g l$^{-1}$) and 50 ml solution of initial Cd(II) concentration of 200 mg l$^{-1}$ were placed in a conical flask, then pH value was modulated to 3, 4, 5, 6, 7 and 8, respectively, oscillated at 180 r.p.m. under 25°C, and finally filtered for examination.

*Effect of reaction temperature on adsorption*. MCFN (0.2 g) was weighed accurately and placed in a conical flask with 50 ml solution of Cd(II) concentration of 200 mg l$^{-1}$, and pH value was modulated to 6. The solution was then oscillated for 180 min at 25, 35, 45 and 55°C to study the Cd(II) adsorption.

All the adsorbed solutions were centrifuged at the rate of 3500 r.p.m. for 20 min and Whatman's 20 μm aperture medium-speed 202 quantitative filter paper was used (GB/T1914-2007), and the Cd(II) mass concentration of the filtrate was determined by flame atomic absorption method [21,30–33]. All measurements were done in triplicate.

## 2.3. Characterization method

*SEM observation*. The morphology and microstructures of MCFN were illustrated by using a field emission SEM. The samples for SEM measurement were mixed with conducting resin in an ultrasonic apparatus and superimposed on an appropriate grid of 3 mm in diameter for the observation.

*FTIR analysis*. The oxygen-containing function groups of MCFN were characterized by using FTIR spectroscopy in pressed KBr (spectroscopic grade) pellets.

## 2.4. Data processing

All the adsorbed solutions were filtered, the supernatant was analysed to measure the mass concentration of Cd(II) ($\rho_e$) and the adsorption capacity and removal rate can be given by equations (2.1) and (2.2), respectively

$$q = \frac{V_0(\rho_0 - \rho_e)}{m} \tag{2.1}$$

$$\eta = \frac{\rho_0 - \rho_e}{\rho_0} \times 100\% , \tag{2.2}$$

where $\rho_0$ and $\rho_e$ are the mass concentration of Cd(II) stock solution and Cd(II) balanced solution, respectively (mg l$^{-1}$), $V_0$ is the volume of the removed Cd(II) solution (l) and $m$ is the mass of ceramic functional nanomaterials (g).

The data of adsorption kinetics are fitted by pseudo-first-order and pseudo-second-order kinetic [34,35], double constant, Elovich and parabolic diffusion models [36,37], and their linear formulae can

be described by equations (2.3), (2.4), (2.5), (2.6) and (2.7), respectively

$$q_t = A - A\exp(-Bt), \tag{2.3}$$

$$\frac{1}{q_t} = A + \frac{B}{t}, \tag{2.4}$$

$$\ln q_t = A + B\ln t, \tag{2.5}$$

$$q_t = A + B\ln t \tag{2.6}$$

and

$$q_t = A + Bt^{0.5}, \tag{2.7}$$

where $q_t$ is the adsorption capacity over time ($t$) (mg g$^{-1}$), $t$ is time (min), and $A$ and $B$ are model parameters. Determination coefficient ($R^2$), $\chi^2$ value ($\varepsilon^2$) and standard error (s.e.) were used to comprehensively test the pros and cons of individual models.

Langmuir [38], Freundlich [39] and Temkin & Pyzhev [40] models were used to fit the isothermal adsorption of Cd(II) on MCFN, and their linear forms can be described by the following equations:

$$\frac{\rho_e}{q} = \frac{\rho_e}{q_m} + \frac{1}{q_m k_L}, \tag{2.8}$$

$$\lg q = \lg k_F + \frac{1}{n}\lg\rho_e \tag{2.9}$$

and

$$q = a\ln\rho_e + b, \tag{2.10}$$

where $q$ is the adsorbing capacity (mg g$^{-1}$), $q_m$ is the maximum adsorbing capacity (mg g$^{-1}$), and $k_L$, $k_F$, $n$, $a$ and $b$ are adsorption constants.

The thermodynamic parameters including Gibbs free energy ($\Delta G$), enthalpy ($\Delta H$) and entropy changes ($\Delta S$) can be described by the following equations [36]:

$$K_0 = \frac{q_e}{c_e}, \tag{2.11}$$

$$\Delta G = -RT\ln K_0 \tag{2.12}$$

and

$$\Delta G = \Delta H - T\Delta S, \tag{2.13}$$

where $R$ is the universal gas constant (8.314 J mol$^{-1}$ K$^{-1}$), $T$ is the temperature (K) and $K_0$ is the distribution coefficient.

# 3. Results and discussion

## 3.1. Kinetic behaviour of Cd(II) adsorption on MCFN

The effect of time on the MCFN adsorption of Cd is shown in figure 1. As can be seen, the adsorption rate of Cd(II) by MCFN was significant during 0–60 min with a large linear slope. However, the adsorption rate slowed down after 60 min which might be due to the saturation of the adsorption sites on the surface of MCFN by Cd(II) and an increase in the resistance of diffusion of free ions to the inner surface. An equilibrium obtained after 180 min with the adsorption capacity of 48.73 mg g$^{-1}$ and the removal rate of 97.46%. These results are similar to the findings of Guo *et al*. [41] and Huang *et al*. [19].

The results of fitting the relationship between the amount and time of Cd(II) adsorption are shown in figure 2, and the fitting parameters are summarized in table 1.

From figure 2, the data were better fitted with pseudo-second-order kinetic and pseudo-first-order kinetic models. $R^2$ (0.99) and $\varepsilon^2$ (0.06) values indicated that pseudo-second-order kinetic was optimal model (table 1). However, the fitting results of Elovich and double constant models were poor, while that of parabolic diffusion model was the worst. The pseudo-second-order kinetic model is based on the assumption that the whole adsorption process is controlled by the chemical adsorption, and it can precisely describe the dynamics of the adsorption process [30], whereas the parabolic diffusion model is based on the assumption that adsorption is controlled by multiple diffusion mechanisms, which is more consistent with the kinetics of particle internal diffusion [42,43]. These are in agreement with the adsorption kinetics studied by Liu *et al*. [44].

## 3.2. Isothermal adsorption of Cd(II) by MCFN

The isothermal adsorption curve of Cd(II) by MCFN is shown in figure 3. As can be seen from figure 3, when the mass concentration of Cd(II) in equilibrium solution was low, the adsorption of Cd(II) by

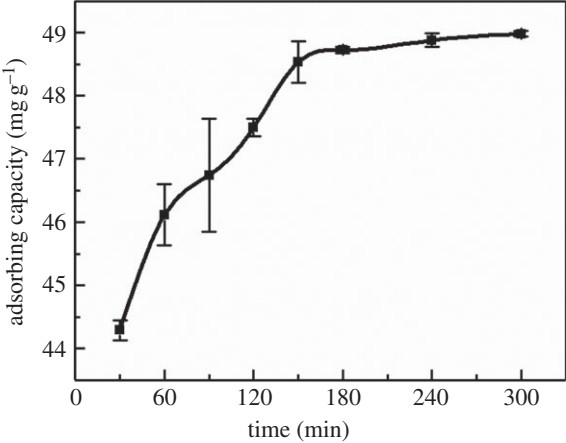

**Figure 1.** The effect of time on Cd(II) adsorption by MCFN (temperature $= 25°$C, pH $= 6$, oscillation rate $= 180$ r.p.m., dosage $=$ 0.2 g).

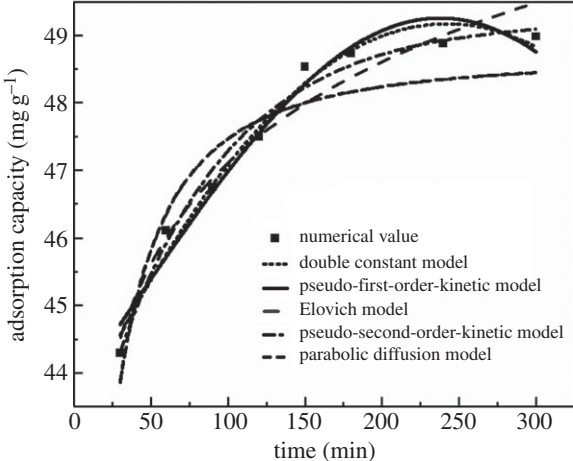

**Figure 2.** The kinetic curves of Cd(II) adsorption by MCFN (temperature $= 25°$C, pH $= 6$, oscillation rate $= 180$ r.p.m., dosage $=$ 0.2 g).

MCFN enhanced rapidly with the increase in the equilibrium mass concentration of Cd(II). Hence, the number and the mass of Cd(II) approaching and absorbing by, respectively, MCFN increased per unit. Then, after the rate of adsorption reached a certain value, MCFN adsorption sites gradually saturated [33,45]. These results are consistent with the findings of Liu *et al.* [46].

The test data were fitted by isothermal adsorption model, and the results are shown in table 2. It can be seen from table 2 that all the three adsorption models could well describe the characteristics of Cd(II) adsorption by MCFN. Comparing determination coefficients ($R^2$) and standard errors of the three models indicated that the Langmuir model was superior. The maximum adsorption capacity of MCFN for Cd(II) at pH 6 and 25°C was 97.09 mg g$^{-1}$.

## 3.3. Influencing factors on Cd(II) adsorption by MCFN

### 3.3.1. The effect of adsorbent dosage on Cd(II) adsorption

The effect of adsorbent dosage on Cd(II) adsorption is shown in figure 4. The removal rate increased slowly and stabilized to 99.98% when the dosage of MCFN was greater than 20 g l$^{-1}$. However, the adsorption capacity gradually decreased from 144.85 to 9.99 mg g$^{-1}$, which might be due to the little addition of MCFN resulting in smaller total specific surface area of the adsorbents and less Cd(II) removal. The active adsorbable spots on the surface and the removal rate of Cd(II) increased with increasing MCFN dosage. The adsorption reached equilibrium when the dosage of MCFN increased to a fixed extent and some adsorption sites were not fully used, thus, resulting in the decrease in

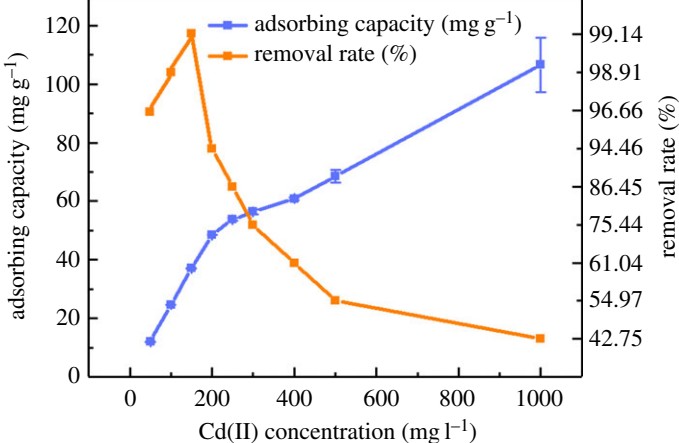

**Figure 3.** The isothermal adsorption curve of Cd(II) by MCFN (temperature = 25°C, pH = 6, oscillation rate = 180 r.p.m., dosage = 0.2 g, contact time = 180 min).

**Table 1.** Fitting parameters of five kinetic models for Cd(II) adsorption by MCFN.

| parameters | pseudo-first-order kinetic model | pseudo-second-order kinetic model | double constant model | Elovich model | parabolic diffusion model |
|---|---|---|---|---|---|
| $A$ | 0.0465 | 49.7512 | 3.6398 | 37.18 | 42.885 |
| $B$ | 11.2414 | 0.0043 | 0.0416 | 2.1568 | 0.3698 |
| $R^2$ | 0.9788 | 0.9999 | 0.9588 | 0.9605 | 0.8865 |
| $\varepsilon^2$ | 0.0978 | 0.0612 | 0.1557 | 0.2049 | 0.4445 |

Cd(II) adsorption. The optimum dosage of adsorbent was $20\,\mathrm{g\,l^{-1}}$ when the adsorption capacity and Cd(II) removal rate were considered comprehensively.

### 3.3.2. The effect of pH value of solution on Cd(II) adsorption

The changes in adsorption properties of adsorbent at initial pH values of 3.0–8.0 were investigated and the results are shown in figure 5. It can be seen that pH value had a great influence on Cd(II) absorption by MCFN. The adsorption capacity of MCFN increased with increasing pH value between 3 and 6.

The optimal pH reaction was determined by the actual material. It is known that pH may affect the adsorption of cadmium by influencing the hydrolysis of Cd(II), the exchange between Cd(II) and $H^+$, the type of adsorptive complexing surface, the charge on adsorptive surface and the distribution coefficient of cadmium in the competitive system [47]. The competitive adsorption of $H^+$ and Cd(II) occurred in the solution under the low pH value and the hydrated hydrogen occupied the adsorption sites resulting in smaller adsorption capacity. However, with increasing pH value, the competitive adsorption effect of $H^+$ decreased, and the negative charge on the adsorbent surface increased, which weakened the electrostatic repulsion of the absorbent surface to Cd(II), while the adsorption capacity and the removal rate of Cd(II) increased [48]. Metal ions precipitated with $OH^-$ when pH value was greater than 7. Therefore, the optimal pH for MCFN adsorbents was 6.

### 3.3.3. The effect of oscillation rate on Cd(II) adsorption

The extraction effect was also affected by the oscillation rate according to the results of the study. As shown in figure 6, MCFN had the lowest adsorption of Cd(II) when oscillation rate was 100 r.p.m. Furthermore, a significant upward trend was found when the oscillation rate increased from 100 to 150 r.p.m., then the catalytic effect of the oscillation rate on the Cd(II) adsorption reached a peak and gradually flattened out.

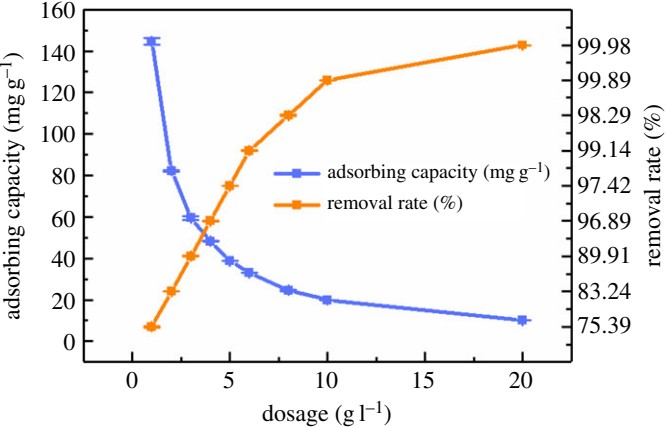

**Figure 4.** The effect of adsorbent dosage on Cd(II) adsorption by MCFN (temperature = 25°C, concentration = 200 mg l$^{-1}$, pH = 6, oscillation rate = 180 r.p.m., contact time = 180 min).

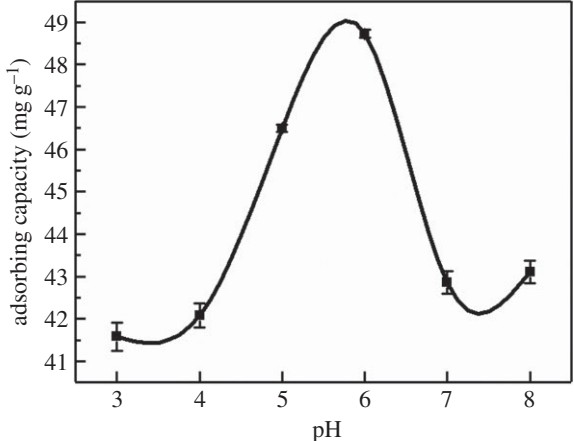

**Figure 5.** The effect of solution pH on Cd(II) adsorption by MCFN (temperature = 25°C, concentration = 200 mg l$^{-1}$, dosage = 0.2 g, oscillation rate = 180 r.p.m., contact time = 180 min).

**Table 2.** Fitting parameters of three isothermal adsorption models for Cd(II) adsorption by MCFN.

| | Langmuir model | | | | Freundlich model | | | | Temkin model | | | |
|---|---|---|---|---|---|---|---|---|---|---|---|---|
| $q_m$/ (mg g$^{-1}$) | $k_L$/ (l mg$^{-1}$) | $R^2$ | s.e. | $K_F$ | $1/n$ | $R^2$ | s.e. | $a$ | $b$ | $R^2$ | s.e. |
| 97.09 | 0.042 | 0.983 | 0.270 | 27.22 | 0.187 | 0.838 | 0.214 | 44.95 | −124.14 | 0.983 | 2.685 |

### 3.3.4. The effect of temperature on Cd(II) adsorption

The temperature range used in this study varied from 25°C to 55°C. It can be seen from figure 7 that the amount of Cd(II) adsorbed by MCFN gradually increased with increasing temperature and the adsorption capacity of Cd(II) tended to be flat when the temperature was higher than 45°C. The results showed that the reaction temperature had a certain effect on the adsorption efficiency of MCFN. According to the theory of chemical adsorption, the chemical bond forces of different substances are much greater than the van der Waals forces of the same substance and other molecules in the adsorption process. Therefore, the adsorption trap is deeper and the action distance is shorter. It is necessary to provide certain energy to cross the energy barrier and to meet the conditions of chemical adsorption [49]. Increasing the reaction temperature is also a way to promote the two substances to cross the chemisorptive energy barrier. When the promoting effect of temperature

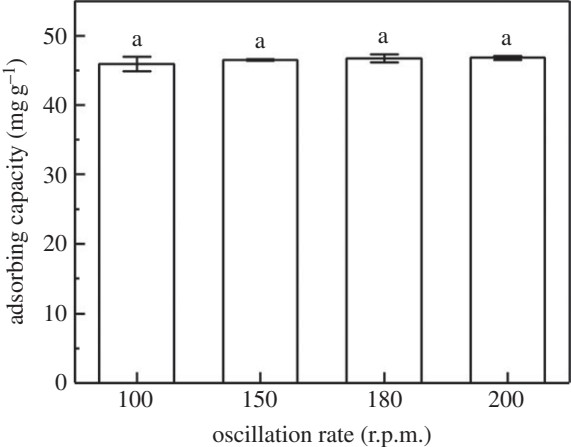

**Figure 6.** The effect of oscillation rate on Cd(II) adsorption by MCFN (temperature = 25°C, concentration = 200 mg l$^{-1}$, dosage = 0.2 g, pH = 6, contact time = 180 min).

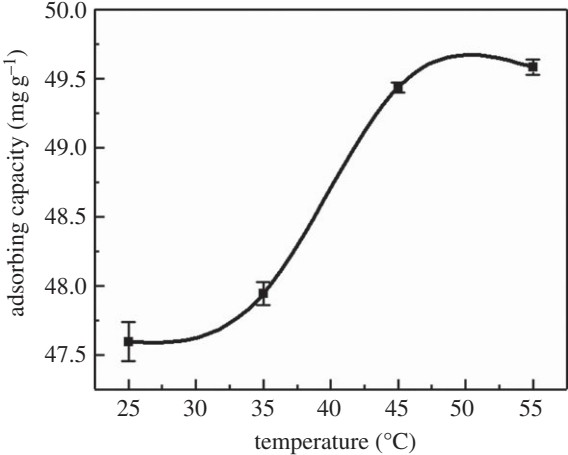

**Figure 7.** The effect of temperature on Cd(II) adsorption (concentration = 200 mg l$^{-1}$, pH = 6, dosage = 0.2 g, oscillation rate = 180 r.p.m., contact time = 180 min).

reached the peak value, the modification effect tended to be gentle. These are consistent with the regularity of modified zeolite researched by Yang *et al.* [50].

The results of the thermodynamic parameters are shown in table 3. The negative $\Delta G$ values indicated that the adsorption of Cd(II) ions onto MCFN was feasible and spontaneous. The positive values of $\Delta H$ suggested the endothermic nature of adsorption, while the positive values of $\Delta S$ revealed the increasing randomness at the solid–solution interface during the adsorption process.

## 3.4. Characterization of the adsorbent

Figure 8 shows the SEM photos before and after Cd(II) adsorption by MCFN. Comparing the surface characteristics of MCFN before and after adsorption, it can be seen that the surface of MCFN was mainly needle-shaped, club-shaped and columnar-shaped before adsorption, while the material was round in shape. There were more spherical and round granular materials in the interspace, and the edges became round and fuzzy after adsorption, which is due to the adsorption pollutants adsorbed on the surface of the material [46]. Through observation and analysis, the club-shaped micropores were filled and closed and the interstitial structures were greatly reduced after absorption indicating that the needle-columnar-shaped microporous structure played a major role in the adsorption process and complex physical and chemical adsorption processes occurred on the surface of micropores.

Figure 9 shows the FTIR spectra with 4000–400 cm$^{-1}$ spectral width before and after adsorption. It can be seen that the shape of peak did not change before and after adsorption, but the infrared

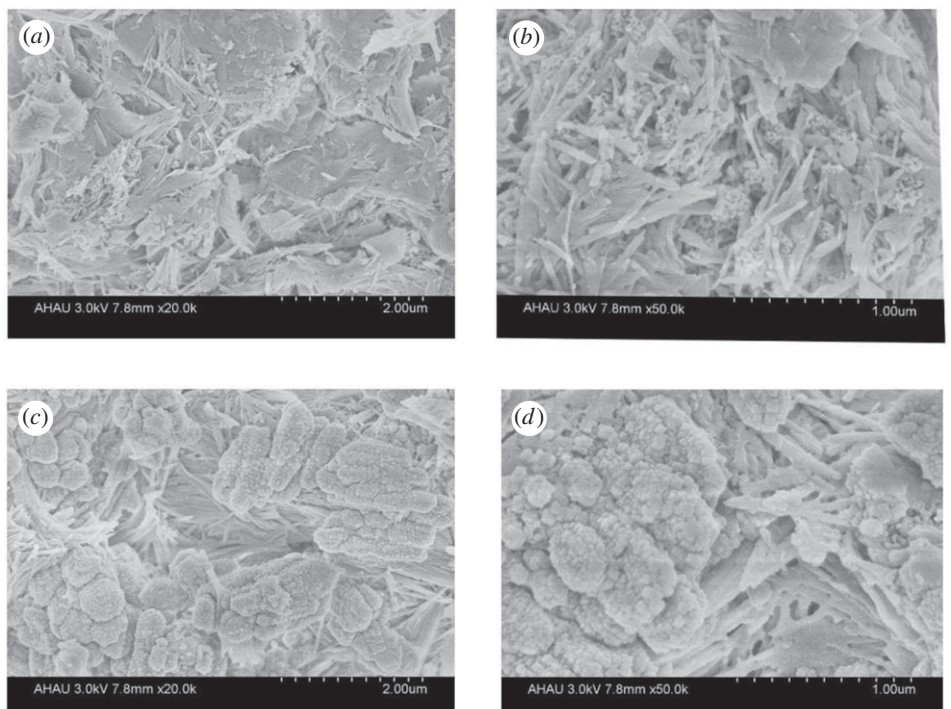

**Figure 8.** The SEM photos before and after Cd(II) adsorption on MCFN. (*a*) The 20 000 times before MCFN adsorption, (*b*) 50 000 times before MCFN adsorption, (*c*) 20 000 times after MCFN adsorption and (*d*) 50 000 times after MCFN adsorption.

**Table 3.** Thermodynamic parameters of Cd(II) adsorption by MCFN at different temperatures.

| temperature (°C) | $K_0$ | $\Delta G$ (kJ mol$^{-1}$) | $\Delta H$ (kJ mol$^{-1}$) | $\Delta S$ (J mol$^{-1}$ K$^{-1}$) |
|---|---|---|---|---|
| 25 | 9.38 | −8.90 | −0.83 | |
| 45 | 28.02 | −9.49 | −0.88 | 27.09 |
| 55 | 43.82 | −9.79 | −0.91 | |

transmittance of four stretching vibration peaks slightly decreased to the low wavenumber. The stretching vibration peaks of intermolecular O–H hydrogen bond or amide lay on 3442 cm$^{-1}$, the stretching vibration of C=O amide was at 1636 cm$^{-1}$, the C–N expansion of amide in 1420–1400 cm$^{-1}$ lay on 1421 cm$^{-1}$, the aromatic compound in out-plane bending vibration region of C–H (1000–650 cm$^{-1}$) in the fingerprint area lay on 798 cm$^{-1}$ and the radical groups were *p*-disubstituted benzene ring or *m*-disubstituted benzene ring. These results indicate that adsorption is a series of chemical reactions produced by radical groups and Cd(II), which is consistent with the inference that chemical adsorption is the main factor in adsorption kinetics [51].

## 4. Conclusion

The adsorption of Cd(II) by MCFN reached to an equilibrium at 180 min. The pseudo-second-order kinetic equation successfully described the adsorbed kinetic process. The isothermal adsorption curve of MCFN conforms to the Langmuir model equation. The optimal dose of MCFN was 20 g l$^{-1}$, because the adsorbent has the best effect in removing Cd from wastewater at this time. The adsorption capacity of MCFN was significantly affected by pH determining the state of Cd(II) in aqueous solution. The maximum adsorption capacity of MCFN for Cd(II) at pH 6, 25°C, dosage of 0.2 g and contact time of 180 min was calculated to be 97.09 mg g$^{-1}$. Likewise to the MCFN dosage, an increase in temperature increased the Cd(II) adsorption. The SEM characterization indicated that the surface of MCFN was rougher and more highly saturated after adsorption with more granular

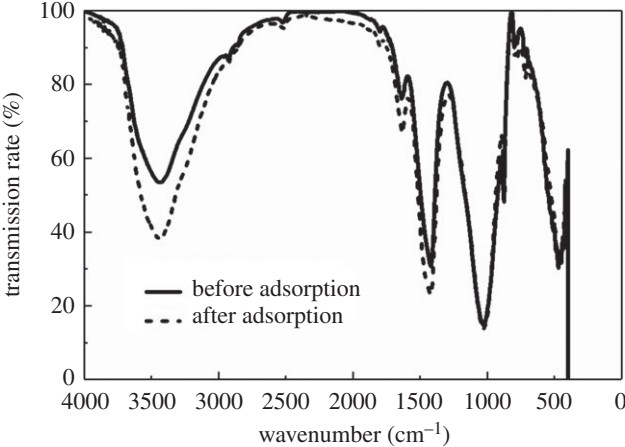

**Figure 9.** The FTIR spectra of MCFN before and after Cd(II) adsorption.

substances and closed pores, suggesting that the adsorption mainly occurs on the surface. The FTIR spectrogram analysis combined with kinetic studies and adsorption thermodynamic parameters revealed that MCFN adsorption of Cd(II) was both physical and chemical. This work demonstrated that MCFN can potentially serve as a promising adsorbent for immobilization of heavy metals in environmental samples.

Data accessibility. Data available from the Dryad Digital Repository at: https://doi.org/10.5061/dryad.sg2637q [52].
Authors' contributions. Z.X. designed the study, carried out the whole laboratory work, participated in data analysis and drafted the manuscript. N.L. and X.W. carried out the statistical analyses and helped drafting the manuscript. H.H. and J.H. put forward key opinions on the revision of the paper giving a great help in data analysis. Y.K.K. interpreted the results and helped to draft the manuscript and revised it. N.X., X.F. and L.Z. helped with data collection, analysis and interpretation of data. All authors gave final approval for publication.
Competing interests. There are no competing interests to declare.
Funding. This work was financially supported by the National Key Research and Development Project (grant no. 2016YFD0801105), National Key Research and Development Program Project (grant no. 2018YFD0800203), Anhui Natural Science Foundation Project (grant no. 1708085MD) and National Natural Science Foundation Project (grant no. 4701575).
Acknowledgements. The authors would like to express their sincere thanks to Prof. Hu, Dr Huang, Dr Kianpoor Kalkhajeh and Dr Ye for their great help. Also, the authors gratefully acknowledge Anhui Province Key Laboratory of Farmland Ecological Conservation and Pollution Prevention.

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
