## [Reviewer comments · Royal Society Open Science]

Review History

RSOS-182195.R0 (Original submission)

Review form: Reviewer 1 (Muhammad Raziq Rahimi Kooh)

Is the manuscript scientifically sound in its present form?

Yes

Are the interpretations and conclusions justified by the results?

Yes

Is the language acceptable?

Yes

Is it clear how to access all supporting data?

Yes

Do you have any ethical concerns with this paper?

No

Have you any concerns about statistical analyses in this paper?

No

Recommendation?

Accept with minor revision (please list in comments)

Comments to the Author(s)

<<SUMMARY OF GENERAL COMMENTS>>

This study investigated the adsorption capability of clay soil on the removal of Cd(II) from aqueous solution. The effects of pH, dosage and contact time were investigated. The adsorbent was characterized by using FTIR and SEM. A total of 3 isotherm and 5 kinetics models were included to characterize the adsorption process.

A routine typical adsorption study. The design of the experiment is fine and the conclusion is supported by the results. However, there are many minor errors/issues that the authors need to work on. Please see my comments below to further improve the quality of the manuscript

<<SPECIFIC COMMENTS>>

1. There are a lot of minor grammatical and spelling errors. Please recheck the whole manuscript. Below are few that are obvious.

pg5line58. The word "and" is missing. Quoting sentence "...by scanning electron microscopy (SEM), Fourier transform infrared spectroscopy (FTIR)"

Pg5line45: "Spectromete"

2. About using abbreviation. Abbreviations that are defined in the abstract will need to be redefined again at first use in the main text. Terms that are already defined should not be redefined again otherwise it is a duplication of effort. Once a term has been defined, avoid switching back to the full term. Please correct.

3. Section 2.3 - there is no need to repeat the model name of the instrument FTIR and SEM because these information were already mentioned in the second paragraph of section 2.1.

4. The isotherm and the kinetics models need to be cited to the work of the original authors. For example:

Pseudo 1st order model to:

S. Lagergren. About the theory of so called adsorption of soluble substances, Kungliga Svenska Vetenskapsakademiens Handlingar, 24, (1898) 1-39.

and Ho's expression of pseudo 2nd order model to:

Y.S. Ho, G. McKay. Sorption of dye from aqueous solution by peat, Chem. Eng. J., 70, (1998) 115-124.

Please do so for the rest of the models.

5. The figure captions need to be informative.

"Fig. 1. The effect of time on Cd(II) adsorption on MCFN" is too vague.

Please include information such as the initial concentration of Cd(II), pH of aqueous solution and adsorbent dosage. Please do so for the rest of the figures.

6. Please adjust the Y-axis scale of Figure 1 to interval of 1.0

7. Please include error bars in Figures 1,3,4,6 and 7.

Review form: Reviewer 2 (Eid Brima)

Is the manuscript scientifically sound in its present form?

No

Are the interpretations and conclusions justified by the results?

No

Is the language acceptable?

Yes

Is it clear how to access all supporting data?

Yes

Do you have any ethical concerns with this paper?

No

Have you any concerns about statistical analyses in this paper?

No

Recommendation?

Major revision is needed (please make suggestions in comments)

Comments to the Author(s)

In general

1- Precision and accuracy of the results were absent (not reported).

2- Conditions and parameters of the techniques/instrument were not reported.

3- The exact pore size of the MCFN was not reported/measured.

page 1, Line 45-50: The optimum contact time and adsorbent dosage was not mentioned.....

Please write these two optimum parameters in the final paragraph in the abstract.

Complete the following statement (conclusion): The maximum adsorption capacity

of Cd(II) on MCFN calculated from Langmuir model at pH 6 and 25 °C was 97.09 (mg/g)

and.....

page 4, Line 40:diameter has the advantage to effectively adsorb and heavy metals. Correct todiameter has the advantage to effectively adsorb heavy metals.

page 6, Line 23: Isothermal adsorption test: 4 (g/L).....etc. Why did you use 4g/L?

page 6, Line 45:0.1 g, 0.15 g, 0.2 g, 0.25 g, 0.3 g, 0.4 g, 0.5 g, and 1.0 g MCFN was added to the solution.....

Still there is no justification of using 4g/L the above mentioned statement was up to 1g..

Fig.4 page 16: Fig. 4. The effect of adsorbent dosage on Cd(II) adsorption on MCFN. This figure showed up to 1g adsorbent dose. Here, again we see a contradiction to your statement of using 4g/L?!!!! please give justification of using such adsorbent dosage.

Fig.8 page 23-25:removed Cd(II) solution, L, m is the quality of ceramic functional nanomaterials, g. This sentence should be corrected: removed Cd(II) solution (L), m is the mass of ceramic functional nanomaterials (g).

Fig.9 page 51-54: Please justify this statement.....the mechanism of action is mainly physical adsorption and ion exchange.

Fig.11 page 53: correct Fsig. 2, to Fig.2

page15, Line 26-35: The effect of adsorbent dosage on Cd(II) adsorption is shown in Fig. 4.

The removal rate of Cd(II) of MCFN gradually increases with the rise of adsorbent dosage, and it increases rapidly from 75.39% to 96.89% between 1.0 (g/L) and 4.0 (g/L)?!

In the above paragraph the last phrase between 1.0 (g/L) and 4.0 (g/L) was not proved on Fig 4. Fig. 4. The effect of adsorbent dosage on Cd(II) adsorption on MCFN, this Figure showed removal up to 100% by using 1g, but no 4g dose appeared on the mentioned figure!!! Please draw another figure to confirm your statement (between 1.0 (g/L) and 4.0 (g/L) or just delete this statement.

page16, Line 7-12: The optimum dosage of the adsorbent is 4 (g/L) when the adsorbing capacity and removal rate of Cd(II) were considered comprehensively.???
This statement is not confirmed and wasn't showed on a Figure? Please give more clarification by drawing a convincing new figure. OR just change the optimum dosage to 1 g in the whole manuscript because that was confirmed and showed on Fig. 4.

page17, Line 31: section: 3.3.4 The effect of temperature on Cd(II) adsorption

- You need to determine thermodynamic parameters such as ΔG , ΔH and ΔS
- ΔG to differentiate between physical or chemical adsorption.
- ΔH to confirm the exothermic or endothermic reaction
- ΔS is the adsorption process spontaneous or non-spontaneous.

Decision letter (RSOS-182195.R0)

08-Feb-2019

Dear Mr Xue:

Title: Study on the Adsorption of Cd(II) in Water by Mesoporous Ceramic Functional Nanomaterials

Manuscript ID: RSOS-182195

The editor assigned to your manuscript has now received comments from reviewers. We would like you to revise your paper in accordance with the referee and Subject Editor suggestions which can be found below (not including confidential reports to the Editor). Please note this decision does not guarantee eventual acceptance.

Please submit your revised paper before 03-Mar-2019. Please note that the revision deadline will expire at 00.00am on this date. If we do not hear from you within this time then it will be assumed that the paper has been withdrawn. In exceptional circumstances, extensions may be possible if agreed with the Editorial Office in advance. We do not allow multiple rounds of revision so we urge you to make every effort to fully address all of the comments at this stage. If deemed necessary by the Editors, your manuscript will be sent back to one or more of the original reviewers for assessment. If the original reviewers are not available we may invite new reviewers.

RSC Associate Editor:
Comments to the Author:
(There are no comments.)

RSC Subject Editor:
Comments to the Author:
(There are no comments.)

Reviewers' Comments to Author:
Reviewer: 1

Comments to the Author(s)

<<SUMMARY OF GENERAL COMMENTS>>

This study investigated the adsorption capability of clay soil on the removal of Cd(II) from aqueous solution. The effects of pH, dosage and contact time were investigated. The adsorbent was characterized by using FTIR and SEM. A total of 3 isotherm and 5 kinetics models were included to characterize the adsorption process.

A routine typical adsorption study. The design of the experiment is fine and the conclusion is supported by the results. However, there are many minor errors/issues that the authors need to work on. Please see my comments below to further improve the quality of the manuscript

<<SPECIFIC COMMENTS>>

1. There are a lot of minor grammatical and spelling errors. Please recheck the whole manuscript. Below are few that are obvious.

pg5line58. The word "and" is missing. Quoting sentence "...byscanning electron microscopy (SEM), Fourier transform infrared spectroscopy (FTIR)"
Pg5line45: "Spectromete"

2. About using abbreviation. Abbreviations that are defined in the abstract will need to be redefined again at first use in the main text. Terms that are already defined should not be redefined again otherwise it is a duplication of effort. Once a term has been defined, avoid switching back to the full term. Please correct.

3. Section 2.3 - there is no need to repeat the model name of the instrument FTIR and SEM because these information were already mentioned in the second paragraph of section 2.1.

4. The isotherm and the kinetics models need to be cited to the work of the original authors.

For example:

Pseudo 1st order model to:

S. Lagergren. About the theory of so called adsorption of soluble substances, *Kungliga Svenska Vetenskapsakademiens Handlingar*, 24, (1898) 1-39.

and Ho's expression of pseudo 2nd order model to:

Y.S. Ho, G. McKay. Sorption of dye from aqueous solution by peat, *Chem. Eng. J.*, 70, (1998) 115-124.

Please do so for the rest of the models.

5. The figure captions need to be informative.

"Fig. 1. The effect of time on Cd(II) adsorption on MCFN" is too vague.

Please include information such as the initial concentration of Cd(II), pH of aqueous solution and adsorbent dosage. Please do so for the rest of the figures.

6. Please adjust the Y-axis scale of Figure 1 to interval of 1.0

7. Please include error bars in Figures 1,3,4,6 and 7.

Reviewer: 2

Comments to the Author(s)

In general

1- Precision and accuracy of the results were absent (not reported).

2- Conditions and parameters of the techniques/instrument were not reported.

3- The exact pore size of the MCFN was not reported/measured.

page 1, Line 45-50: The optimum contact time and adsorbent dosage was not mentioned.....

Please write these two optimum parameters in the final paragraph in the abstract.

Complete the following statement (conclusion): The maximum adsorption capacity of Cd(II) on MCFN calculated from Langmuir model at pH 6 and 25 °C was 97.09 (mg/g) and.....

page 4, Line 40:diameter has the advantage to effectively adsorb and heavy metals. Correct todiameter has the advantage to effectively adsorb heavy metals.

page 6, Line 23: Isothermal adsorption test: 4 (g/L).....etc. Why did you use 4g/L?

page 6, Line 45:0.1 g, 0.15 g, 0.2 g, 0.25 g, 0.3 g, 0.4 g, 0.5 g, and 1.0 g MCFN was added to the solution.....

Still there is no justification of using 4g/L the above mentioned statement was up to 1g..

Fig.4 page 16: Fig. 4. The effect of adsorbent dosage on Cd(II) adsorption on MCFN. This figure showed up to 1g adsorbent dose. Here, again we see a contradiction to your statement of using 4g/L?!! please give justification of using such adsorbent dosage.

Fig.8 page 23-25:removed Cd(II) solution, L, m is the quality of ceramic functional nanomaterials, g. This sentence should be corrected: removed Cd(II) solution (L), m is the mass of ceramic functional nanomaterials (g).

Fig.9 page 51-54: Please justify this statement.....the mechanism of action is mainly physical adsorption and ion exchange.

Fig.11 page 53: correct Fsig. 2, to Fig.2

page15, Line 26-35: The effect of adsorbent dosage on Cd(II) adsorption is shown in Fig. 4.

The removal rate of Cd(II) of MCFN gradually increases with the rise of adsorbent dosage, and it increases rapidly from 75.39% to 96.89% between 1.0 (g/L) and 4.0 (g/L)?!

In the above paragraph the last phrase between 1.0 (g/L) and 4.0 (g/L) was not proved on Fig 4.

Fig. 4. The effect of adsorbent dosage on Cd(II) adsorption on MCFN, this Figure showed removal up to 100% by using 1g, but no 4g dose appeared on the mentioned figure!!! Please draw another figure to confirm your statement (between 1.0 (g/L) and 4.0 (g/L) or just delete this statement.

page16, Line 7-12: The optimum dosage of the adsorbent is 4 (g/L) when the adsorbing capacity and removal rate of Cd(II) were considered comprehensively.???

This statement is not confirmed and wasn't showed on a Figure? Please give more clarification by drawing a convincing new figure. OR just change the optimum dosage to 1 g in the whole manuscript because that was confirmed and showed on Fig. 4.

page17, Line 31: section: 3.3.4 The effect of temperature on Cd(II) adsorption

- You need to determine thermodynamic parameters such as ΔG , ΔH and ΔS
- ΔG to differentiate between physical or chemical adsorption.
- ΔH to confirm the exothermic or endothermic reaction
- ΔS is the adsorption process spontaneous or non-spontaneous.

Author's Response to Decision Letter for (RSOS-182195.R0)

See Appendix A.

Decision letter (RSOS-182195.R1)

19-Mar-2019

Dear Mr Xue:

Title: Adsorption of Cd(II) in Water by Mesoporous Ceramic Functional Nanomaterials
Manuscript ID: RSOS-182195.R1

It is a pleasure to accept your manuscript in its current form for publication in Royal Society Open Science. The chemistry content of Royal Society Open Science is published in collaboration with the Royal Society of Chemistry.

RSC Associate Editor
Comments to the Author:
(There are no comments.)

Reviewer(s)' Comments to Author:

Appendix A

March 6, 2019

Dear Dr Laura Smith,

Thank you very much for your letter and the comments from the referees about our paper submitted to Royal Society Open Science (Manuscript ID RSOS-182195). We have checked the manuscript and revised it according to the comments. We submit here the revised manuscript as well as a list of changes.

Response to Reviewer 1:

Thanks for your comments on our paper. We have revised our paper according to your comments:

- 1) First, we checked and revised the grammar and spelling.
- 2) And check the definition abbreviations to avoid duplication.
- 3) Then, according to your suggestion, the isotherms, kinetic models and adsorption thermodynamic parameters are referenced to the original author's work.
- 4) We add error bars to the image of the article and give basic information.
- 5) And add basic information about the test, such as pH, initial concentration, dosage, temperature, etc., to all graphics as you suggest.

Response to Reviewer 2:

Many thanks for your comments on our paper. We have revised our paper according to your comments:

- 1) Based on your suggestion, we provide accurate experimental parameter reports and add relevant data to explain the problem.
- 2) We supplement the optimum contact time (180 min) and adsorbent dosage (1 g) and explain the experimental conditions.
- 3) For the dosage of adsorbent used, the situation is as follows: because the experimental treatment is to add 0.1g, 0.15g, 0.2g, 0.25g, 0.3g, 0.4g, 0.5g and 1.0g MCFN into 50 mL Cd²⁺ concentration of 200 (mg/L) CdCl₂ solution, because we add 0.2 g to 50 mL, and it was convert to 4 (g/L). However, from the adsorption efficiency of the experimental results, when the dosage is 1g, the adsorption rate is the highest and the effect is the best. Your opinion is correct. So we changed to 1g, convert it to 20 (g/L).
- 4) For the questions you have asked about physical adsorption and ion exchange, we remove this statement and add the adsorption thermodynamic parameters (ΔG , ΔH and ΔS) to the adsorption principle as per your opinion. Thermodynamic parameters (ΔG , ΔH and ΔS) can well explain the adsorption properties.
- 5) Finally, we have corrected the figures, spelling and details that you pointed out, and carefully checked the others and revised and improved the contents.

Thank you very much for the excellent and professional revision of our manuscript. Overall the comments have been fair, encouraging and constructive. We have learned much from it.

If you have any question about this paper, please don't hesitate to let me know.

Sincerely yours,

Hongxiang Hu